# Post-Developmental Roles of Notch Signaling in the Nervous System

**DOI:** 10.3390/biom10070985

**Published:** 2020-07-01

**Authors:** Jose L. Salazar, Sheng-An Yang, Shinya Yamamoto

**Affiliations:** 1Department of Molecular and Human Genetics, Baylor College of Medicine (BCM), Houston, TX 77030, USA; jose.salazar@bcm.edu (J.L.S.); sheng-an.yang@bcm.edu (S.-A.Y.); 2Department of Neuroscience, BCM, Houston, TX 77030, USA; 3Program in Developmental Biology, BCM, Houston, TX 77030, USA; 4Development, Disease Models & Therapeutics Graduate Program, BCM, Houston, TX 77030, USA; 5Jan and Dan Duncan Neurological Research Institute, Texas Children’s Hospital, Houston, TX 77030, USA

**Keywords:** *Drosophila melanogaster*, Notch signaling, adult brain, neuropsychiatric diseases, neurodegeneration

## Abstract

Since its discovery in *Drosophila*, the Notch signaling pathway has been studied in numerous developmental contexts in diverse multicellular organisms. The role of Notch signaling in nervous system development has been extensively investigated by numerous scientists, partially because many of the core Notch signaling components were initially identified through their dramatic ‘neurogenic’ phenotype of developing fruit fly embryos. Components of the Notch signaling pathway continue to be expressed in mature neurons and glia cells, which is suggestive of a role in the post-developmental nervous system. The Notch pathway has been, so far, implicated in learning and memory, social behavior, addiction, and other complex behaviors using genetic model organisms including *Drosophila* and mice. Additionally, Notch signaling has been shown to play a modulatory role in several neurodegenerative disease model animals and in mediating neural toxicity of several environmental factors. In this paper, we summarize the knowledge pertaining to the post-developmental roles of Notch signaling in the nervous system with a focus on discoveries made using the fruit fly as a model system as well as relevant studies in *C elegans*, mouse, rat, and cellular models. Since components of this pathway have been implicated in the pathogenesis of numerous psychiatric and neurodegenerative disorders in human, understanding the role of Notch signaling in the mature brain using model organisms will likely provide novel insights into the mechanisms underlying these diseases.

## 1. Introduction

Notch signaling is an evolutionarily conserved signaling pathway that has primarily been studied in the context of development and cancer [1,2,3]. Canonical Notch signaling is mediated by Notch receptors [*NOTCH1-4* in human, *Notch* (*N*) in flies] that are activated by the Delta-family [*DLL1*, *DLL3* and *DLL4* in human, *Delta* (*Dl*) in flies] or Serrate/Jagged-family [*JAG1* and *JAG2* in human, *Serrate* (*Ser*) in flies] ligands presented by neighboring cells [4,5]. Following ligand-receptor interactions, a series of proteolytic cleavage events mediated by ADAM10 (A Disintegrin And Metalloprotease 10) and the γ-secretase complex release the intracellular domain of Notch receptors (NICD) from the membrane. NICD then translocates to the nucleus and binds to CSL (CBF-1, Suppressor of Hairless, Lag-1)-family DNA binding transcription factors, Su(H) (Suppressor of Hairless) in *Drosophila*, and RBPJ (Recombination signal Binding Protein for immunoglobulin kappa J region) in mammals. Mastermind-family coactivators are also recruited to facilitate the transcription of Notch downstream target genes such as *HES* [*Hairy and Enhancer of Split* (*E(spl)*)] family genes [4,5]. While genes that encode the core components of this pathway were cloned in the 1980s and 1990s, many subsequent studies identified hundreds of additional genes that fine-tune this pathway in a context-specific manner [6]. Furthermore, several non-canonical modes of Notch signaling that involve ligand-independent activation or transcriptional events that are independent of CSL have been reported, highlighting the complexity of the pathway [7,8,9]. While rare inherited or de novo variants in genes that encode core components of this pathway have been linked to a number of genetic disorders, most of which are congenital, somatic mutations in a similar set of genes are often found in diverse types of leukemia and solid tumors [10,11,12].

The role of Notch signaling in nervous system development has been extensively investigated because this was the first organ system in which this pathway was shown to play a fundamental role. Since the pioneering discovery by Poulson in the 1930s that Notch signaling is required to divide the developing ectoderm into neural and non-neural tissues, many studies further identified that this pathway is reiteratively used to make diverse developmental decisions in the nervous system [13]. This includes proliferation and maintenance of neural stem cells, specification of fates of their daughter cells, cell migration, differentiation, and cell death [14,15]. Since the initial reports in the mid-1990s that elucidated Notch receptors are expressed in terminally differentiated neurons [16,17], researchers have been exploring the post-developmental function of Notch signaling in the nervous system. In addition to being implicated in the pathogenesis of neurodevelopmental disorders including Down syndrome, intellectual/developmental disability, and autism spectrum disorders [18,19,20,21,22,23,24,25,26,27], recent studies have suggested the involvement of Notch signaling in diverse neuropsychiatric diseases including schizophrenia [28,29], bipolar disorder [30,31], and major depressive disorder [32,33] in humans.

In this manuscript, we review the studies that have probed the role of Notch signaling in post-developmental neurons and glia cells to summarize the current understanding regarding the function of this pathway in neural physiology and maintenance. We primarily focus on studies using the fruit fly *Drosophila melanogaster* and refer to related experiments performed in mammalian and other model systems whenever appropriate. Although exciting research related to the role of Notch signaling in adult neurogenesis as well as non-neuronal cells in the brain, including blood vascular and immune components, are actively being studied by many experts in the field, we will only briefly touch upon these topics and, instead, refer the readers to the following review articles that cover these topics in more detail [34,35,36,37]. This is because there are very limited reports of adult neurogenesis in the mature fly brain [38], and vertebrates and invertebrates have some differences in the circulatory (e.g., flies have an open circulatory system and rely on their trachea for respiration) and immune (e.g., although cells that have microglia-like function exist, flies lack adaptive immune cells) systems [39,40].

## 2. Post-Developmental Notch Signaling in Behavior and Neural Physiology (Table 1)

### 2.1. Notch Signaling in Learning and Memory

Learning and memory are fundamental cognitive tasks that are conserved in most species that have a nervous system. ‘Learning’ allows an organism to associate two or more distinct sensory cues and such information is stored in the form of ‘memories.’ These memories allow an individual/animal to adjust its behaviors to maximize their fitness to survive and to reproduce. The first direct evidence that implicated the post-developmental role of Notch signaling in the nervous system was obtained by two groups that performed conditional experiments in *Drosophila* in 2004 [41,42]. Using temperature-sensitive alleles of *Drosophila Notch* (*N^ts1^*, *N^ts2^*), cell type specific RNA interference (RNAi), or through temporally controlled expression of a dominant negative form of the Notch receptor (*N^Δcdc10rpts^*), two groups revealed that the *Notch* gene is required to establish stable long-term memory (LTM) [41,42]. Learning and memory in *Drosophila* can be assessed via several assay systems. Presente et al. employed the conditional courtship assay, which takes advantage of an innate behavior that a male fly displays when courting a female fly in an attempt to copulate and reproduce (Figure 1A) [42]. When a male fly encounters a female fly, the male displays a complex repertoire of behaviors referred to as the ‘courtship dance’ [43,44]. If the female fly has not mated with another male fly, these virgin females will accept a male fly to copulate. However, if the female fly is a non-virgin, she strongly rejects the male by decamping, kicking, displaying a wing threat, and by extruding her ovipositor (Figure 1A) [45,46]. This experience decreases the male’s vigor in engaging in their next courtship, even if they are presented with a virgin female that is receptive. The memory formed through this assay is typically maintained for several days in a wild-type fly. Presente et al. found that, although *N^ts1^* flies placed under restrictive conditions post-developmentally were able to learn this task and maintain its memory for a short time-frame (~30 min), their courtship index returned to baseline after two days. These data suggest that, in this context, Notch signaling is not required for learning and the formation of short-term memory (STM), but is necessary for the conversion of STM to LTM. The authors further performed a second learning and memory assay based on a Pavlovian olfactory conditional test [42], which was also performed by Ge et al. using different genetic reagents [41]. In this assay, a group of flies is repetitively exposed to a neutral order (e.g., benzaldehyde) that is temporally paired with series of electric shocks (Figure 1B). During the rest of the intervals, the same flies are exposed to another neutral order (e.g., 3-octanol). These flies are then placed into a T-maze to determine whether they were able to associate the first order with the unpleasant experience caused by the electric shock. Similar to the results from the conditioned courtship assays, post-developmental inactivation of Notch signaling caused a significant reduction in LTM without affecting learning and STM. Ge et al. further showed that transient increase in Notch receptor expression (using a heat-shock promotor driven full length *Notch* transgene) three hours prior to the training session is sufficient to facilitate LTM without affecting the efficacy of learning or STM [41]. These data indicate that Notch signaling is not only necessary for LTM formation but can actively facilitate this process. To further assess the cell types in which Notch signaling is acting to promote LTM, Presente et al. performed cell type specific RNAi experiments using the GAL4/UAS system [42]. They found that UAS-*Notch* RNAi driven by a GAL4 line that is expressed in many neurons of the mushroom body (*c772-GAL4*) was able to recapitulate the phenotype seen in the *N^ts1^* allele. Since the mushroom body of insect brains plays a central role in learning and memory [47,48], Notch signaling is likely playing a cell autonomous role in a subset of neurons that convert STM to LTM. Although one mammalian study has shown that mice that are heterozygous for a constitutive null allele of *Notch1* exhibit learning and memory defects [49], the experiments in flies were the first to unambiguously demonstrate that Notch receptors have a post-developmental role in the adult nervous system.

The function of Notch signaling in LTM formation in *Drosophila* was further assessed by manipulation of the *Su(H)* gene, which is the sole CSL family transcription factor that mediates canonical Notch signaling in *Drosophila* [50]. Song et al. found that Su(H) protein is widely expressed in the *Drosophila* adult brain, and further demonstrated that flies heterozygous for loss of function (LOF) alleles of *Su(H)* [*Su(H)^1B115^* and *Su(H)^HG36^*] have reduced LTM formation. These flies did not show any defects in learning and STM [50], which indicates the importance of downstream target genes of the canonical Notch signaling pathway in this behavior. It is important to note, however, that since this study did not control the timing of the loss of *Su(H)*, it is still possible that the developmental role of Su(H) contributes to the LTM defect reported in this study. The same group also found that transient over-expression of Su(H) using a heat-shock promotor driven transgene can also inhibit LTM formation, which leads to their conclusion that an optimal level of Su(H) is critical. Since Su(H) can act as a transcriptional inhibitor in the absence of active Notch, further transcriptomic studies are required to understand how both reduction or an increase in Su(H) levels may lead to LTM defects. In an independent study, Zhang et al. assessed the impact of ectopically activating Notch signaling by over-expressing NICD using a heat-shock promotor driven transgene [51]. In contrast to the facilitation of LTM that was seen by over-expressing the full-length form of Notch prior to the onset of Pavlovian conditioning, over-expression of NICD using the same paradigm lead to inhibition of LTM formation [52]. Hence, it is likely that an optimum level of Notch activation is required for proper formation of LTM, or that uncontrolled Notch over-activation (which is more likely to happen when over-expressing the NICD form compared to the full-length form) can interfere with proper LTM consolidation.

How does canonical Notch signaling facilitate LTM in *Drosophila*? One study took a hypothesis-driven approach and looked at the effect on CREB (cAMP response element binding protein), a mediator of LTM in *Drosophila*, and other species [53,54]. The critical distinction between STM and LTM is the requirement of de novo transcription and protein synthesis following neuronal excitation [55,56]. For learning and STM formation, neuronal changes that occur following the initial training do not require new protein synthesis, whereas this is essential for LTM formation. CREB is a transcription factor that is activated in response to increases in cAMP (cyclic adenosine monophosphate), and pioneering studies in *Aplysia* has established its role in converting STM to LTM [57]. By using mutant alleles of *Notch* (primarily *N^nd-1^* and *N^nd-3^*), Zhang et al. showed that the phosphorylation status of CREB (CrebB in flies, also known as dCrebB-17A) in adult fly brains is altered upon manipulation of Notch activity [51]. Based on this and other data sets [58], the authors proposed a hypothesis in which Notch regulates CREB phosphorylation through a non-canonical mechanism using Protein Kinase C (PKC, encoded by the fly gene *Pkc98E*), which is a model that requires further validation and mechanistic dissections. Importantly, cross-talks between Notch and CREB signaling pathways have been revealed in other species and systems [59,60], even though the suggested mechanisms of interaction are different in different contexts. In another study, Matsuno et al. identified a cell adhesion molecule that functions downstream of Notch signaling in LTM formation [61]. Through a forward genetic screen using random transposon insertions [62], a mutation named *ruslan* was shown to exhibit an LTM phenotype. This allele was mapped to a gene named *klingon* (*klg*), which is a gene that encodes a homophilic cell adhesion molecule that had previously been studied in the context of photoreceptor development in the eye [61]. The authors found that Klg protein levels are increased during LTM formation in a Notch signaling dependent fashion, and post-developmental restoration of wild-type Klg in *klg* mutant flies was shown to be sufficient to rescue the LTM defect, which demonstrated its post-developmental requirement. Additional genetic epistasis experiments showed that reduction of *klg* can suppress the LTM by facilitating the effect of full-length Notch over-expression. Although the precise function of Klg in LTM formation is still unclear, the same group determined that Klg is expressed in the junction between neurons and glia cells [63], and that it likely functions by modulating glutamate transport between these two cell types [64]. In summary, Notch signaling facilitates LTM in *Drosophila* by altering CREB signaling and also potentially by modulating neuron-glia interactions.

In addition to these studies in *Drosophila* and the pioneering study by Costa et al. using mice [49], several studies in mammals have also explored the role for post-developmental Notch signaling in learning and memory. For example, heterozygous *Jag1* null mutant mice were shown to be defective in spatial memory formation [65], which is a phenotype that may be due to developmental and/or post-developmental roles of this gene in learning and memory. Furthermore, postnatal knockout of *Rbpj* [66] and *Adam10* [67] in the brain using *CamKII-Cre* have also been shown to exhibit defects in learning and memory. Additionally, a study in rats showed that *Notch2* is expressed in the hippocampus and cerebellum, where its expression becomes significantly upregulated upon establishment of long-term spatial memory [68]. Although these parallels suggest that Notch signaling may have already been utilized for primitive forms of learning and memory in the urbilaterian, the last common ancestor of invertebrate and vertebrate species [69], several differences found between fly and mammalian systems are worth mentioning. First, while Notch signaling seems to be primarily required for LTM formation in *Drosophila*, studies in mice and rats suggest that this pathway is also required for learning. This could be due to fundamental differences between how learning and memory is encoded in the brain of flies and mammals, or it could be due to the differences in the type of memory tested in different experiments (i.e., mostly episodic memory in flies versus spatial memory in mammals). Second, Notch signaling plays key roles in adult neurogenesis in the mammalian brain, while there are very few reports of adult neurogenesis reported in the fly brain [38]. Considering that adult neurogenesis in the hippocampus plays important roles in learning and memory in mammals, Notch signaling may have acquired new roles in this process during evolution. Third, the neuronal circuits that mediate learning and memory in flies (e.g., mushroom body) and mammals (e.g., hippocampus, neocortex) are anatomically very different and are not considered to be homologous structures. Despite these differences, studies using *Drosophila* will likely to continue to provide new insights into how Notch signaling regulates learning and memory and other brain functions in other model organisms as well as humans.

### 2.2. Notch Signaling in Reward and Addiction

Learning and memory are tightly coupled to reward and addiction because they converge on a similar set of cell types, molecules, and mechanisms [70,71]. Learning is often associated with reward or punishment to pair positive or negative values to certain stimuli. Training is a process in which this pairing is performed repetitively to reinforce a specific coupling between a sensory cue and a value. Through this process, memory can be formed and facilitated through the action of neuromodulators including serotonin and dopamine. Molecules that are associated with addiction such as recreational drugs, alcohol, and nicotine impinge on these neuromodulator pathways to alter behavior, and the underlying molecular pathway and mechanisms are highly conserved [72,73,74]. Over the past decade, geneticists have been using *Drosophila* to understand the molecular basis of addiction by using sophisticated genetic tools and specialized behavioral assays [75,76,77].

Recent studies are beginning to unravel the role of Notch signaling in alcohol reward and addiction [78,79]. *Drosophila* exhibit a natural preference for low non-intoxicating doses of ethanol (EtOH) because they prefer to lay their eggs in places where there is plenty of yeast (primary food source for adult and larval flies). In contrast, high-concentration of EtOH is initially aversive because it can intoxicate flies in a similar manner in which alcohol intake can intoxicate humans and other mammalian species [80]. Upon exposure of flies to EtOH vapor, these animals eventually develop a preference to this stimulus (Figure 1C) [78]. Even when the authors paired EtOH exposure to electric shocks that provide strong aversive cues to unconditioned flies, conditioned flies still showed a strong EtOH seeking behavior in the presence of this harsh manipulation (i.e., the flies still go for a drink even when they know they are going to be ‘zapped’). Considering that one of the hallmarks of addiction is continued alcohol/drug usage despite negative consequences, one can assert that *Drosophila* can become addicted to EtOH. The same authors further showed that dopaminergic neurons and the mushroom body of the fly brain play critical roles in the formation of this addictive behavior, which, again, shows the tight link between ‘learning and memory’ and ‘reward and addiction.’ In the final section of their paper, the authors performed a genetic screen to identify genes in the mushroom body that are critical for the memory of EtOH reward, and identified *scabrous* (*sca*) as one of the strongest hits [78]. Scabrous is a secreted fibrinogen-related protein that modulates Notch signaling during bristle and eye development in *Drosophila* [81,82]. Although the precise function of Scabrous is still obscure, it can physically bind to Notch to facilitate its signaling activities [83,84,85]. Some genes that are critical for Notch signaling perform their actions by regulating the secretion and distribution of Scabrous [86,87], which suggests that Scabrous is an important node to precisely fine-tune signaling output in certain contexts. In a follow-up study, the same group found that canonical Notch signaling was activated upon repetitive EtOH exposure and that Scabrous potentiates this activity [79]. One of the downstream genes activated in mushroom body neurons following repetitive EtOH exposure by Notch signaling is *Dop2R*, a D2-like dopamine receptor encoding gene. Changes in the human D2 dopamine receptor *DRD2* expression has been linked to addictive behaviors in humans [88,89], which indicates that Notch signaling mediates this addiction-like behavior in flies by modulating an evolutionarily conserved pathway that also governs addiction in humans. Whether Notch signaling play similar roles in facilitating alcohol and other types of addiction in other species requires further investigation.

### 2.3. Notch Signaling in Sleep Homeostasis

The mushroom body of the fly brain not only controls ‘learning and memory’ and ‘reward and addiction’ but also ‘sleep’ [90,91]. Sleep is a universal phenomenon that has been observed in every species with a nervous system that has been investigated so far [92]. Sleep is intimately tied to ‘learning and memory,’ and one of the key functions of sleep is to help organize and consolidate important memories in the brain [93]. A number of criteria distinguishes sleep from a period of minimal movement. For example, animals that are sleeping have increased sensory threshold, which means that their response to external stimuli becomes reduced while they are sleeping. In addition, animals that are sleep deprived show future increases in sleep, which is a phenomena referred to as ‘sleep rebound’ (Figure 1D) [94]. Seugnet et al. identified that Notch signaling is required for this homeostatic response based on the observation that flies with mutations in a gene called *bunched* (*bun*) show defects in sleep rebound induction [95]. The *bunched* gene encodes a transcription factor that is orthologous to human TSC22 domain family proteins, and has been identified to function as a negative regulator of Notch signaling during oogenesis [96,97]. The expression of fly *bunched* as well as mouse and human *TSC22D* transcripts have been found to be upregulated in sleep-deprived individuals, which suggests that the expression of this transcription factor can serve as a biomarker of sleep demand [95]. The authors were able to show that a gain-of-function allele of *Notch* (*N^spl-1^*) or over-expression of the *Delta* ligand in mushroom body neurons was sufficient to abolish sleep rebound, which suggests that hyper-activation of Notch signaling can modulate sleep homeostasis. In the same study, the author showed that Notch is predominantly expressed in glial cells, while Delta is present in neurons in the mushroom body. Glia-specific over-expression of NICD was also sufficient to abolish sleep rebound, which suggests that Delta ligands present on mushroom body neurons regulate sleep by activating Notch signaling in neighboring glial cells [95]. Sleep-deprived animals are known to perform poorly in tasks that involve ‘learning and memory.’ Such a reduction in ‘learning and memory’ can also be suppressed by activating Notch signaling via mushroom body-specific over-expression of Delta or glia-specific over-expression of NICD. Although most of the experiments performed in this paper were based on over-expression experiments that need to be interpreted with caution, this was one of the first studies that linked post-developmental Notch signaling to regulation of sleep.

In the same issue that Seugnet et al. reported the role of Notch signaling in sleep using *Drosophila* [95], Singh et al. published a manuscript reporting a role for Notch signaling in a sleep-like behavior in *C. elegans* called ‘lethargus’ [98]. Lethargus is a quiet behavioral state that occurs before each molting stage and has many sleep-like properties [99]. Through the study of *osm-7* and *osm-11* genes that encode co-ligands that can activate Notch receptors (encoded by *lin-12* and *glp-1*) in *C. elegans*, the authors found that activation of Notch signaling can induce a lethargus-like quiescence state in adult worms (anachronistic quiescence). Additional experiments in larval animals further showed that both decreased and increased Notch signaling affect developmental lethargus. Epistasis experiments showed that Notch signaling works upstream of *egl-4*, which is a gene that encodes a cGMP-dependent kinase previously known to be required for developmental lethargus. The authors of this study further performed modifier screens to identify genes that function downstream of Notch signaling to modulate sleep-like behavior in worms by identifying the importance of specific G proteins in this behavior [100].

In contrast to these studies in invertebrate model organisms with established functional links between sleep/sleep-like behaviors and Notch signaling, the link between sleep and Notch signaling in mammals is still obscure. One study using rats identified that Notch signaling-related genes are found to be down-regulated in the hippocampus of sleep-deprived animals [101]. In another study, a link between Notch signaling and insomnia was proposed through analysis of a large GWAS (genome-wide association study) dataset [102]. Functional studies in rodent models will be required to assess whether Notch signaling affects sleep in mammals, and whether any parallels can be found in invertebrate species.

### 2.4. Post-Developmental Notch Signaling in Other Behaviors

Notch signaling in the post-developmental brain has also been shown to contribute to other aspects of behavior in mammalian species that are more difficult to study in *Drosophila* models. Links between Notch signaling and fear, anxiety and depression-like states induced by chronic stress has been explored in mice by a number of groups [65,103,104,105,106,107,108]. Links between Notch signaling and pain perception in the peripheral nervous system and spinal cord have been explored in rats, primarily through pharmacological approaches [109,110,111,112,113,114,115]. Pharmacological studies in mice, zebrafish, and human cells also suggested a link between opioid and Notch signaling pathways [116,117], which suggests that Notch may regulate pain perception in both the central and peripheral nervous systems. Although the link between Notch signaling and stress has not yet been established in humans, it is interesting to note that one study linked Notch signaling to posttraumatic stress disorder (PTSD) through association studies based on genetic and epigenetic analysis of survivors of war and genocide [118]. A link between Notch signaling and sociability has been proposed based on conditional KO (cKO) studies of the *Hes1* gene in inhibitory neurons of the adult mouse brain [108]. It is worth mentioning that several genes linked to autism spectrum disorders in humans are known regulators of Notch signaling (e.g., *Neurobeachin*) [119,120], which suggests a potential link between autism and Notch signaling. As more behavior assays in rodent are developed that can reflect certain aspects of complicated behaviors in humans, we envision additional links between Notch signaling and other sophisticated behavioral and mental states in humans that will be proposed.

## 3. Mechanisms by Which Post-Developmental Notch Signaling Regulate Behavior (Table 2)

### 3.1. Notch Signaling is Regulated by Neuronal Activity

One of the first experiments indicating that Notch signaling may be regulated by neuronal activity was performed in *Drosophila* using the larval motor system [121]. In this study, de Bivort et al. found that the level of Notch receptor was increased in the brains of particular mutant fly strains that have increased neuronal activity such as *ether a go-go* and *Shaker* double mutants, which lack key K^+^ channel subunits [122], and *dunce* mutants, which are defective for cAMP phosphodiesterase that increases cAMP signaling [123]. Although this study did not examine whether downstream targets of Notch signaling were upregulated in these animals, and the study was performed in animals that were still undergoing development, further studies in adult flies and mice demonstrated that Notch activity can be regulated by neuronal activity. By studying the olfactory system of *Drosophila* using sophisticated Notch reporter constructs, Lieber et al. identified that Notch activity is upregulated by neuronal activity in this context [124,125]. Notch activation occurs in olfactory receptor neurons (ORNs) in response to selective stimuli in a Delta ligand-dependent fashion. Activation of Notch signaling in ORNs not only requires the odorant receptors that perceive the sensory cues but also requires synaptic transmission because the Notch activation can be suppressed by expressing Tetanus toxin, which is an inhibitor of synaptic vesicle release. Although not investigated extensively, Lieber et al. also reported that Notch activation can be seen in sensory neurons that mediate hearing (chordotonal neurons) and taste (gustatory neurons) [125], which suggests that activity-dependent Notch signaling activation is not limited to the olfactory system.

Similarly, mammalian Notch1 receptor activation was found to be under the control of synaptic activity when studying the somatosensory cortex and hippocampus in mice [126]. In this context, Notch1 is found at the synapse together with Jag1, and its activation is dependent on *Arc* (a.k.a. *Arg3.1*), which is a neuronal activity-regulated plasticity gene. Although the precise mechanism is yet unknown, neuronal activation seems to somehow trigger proteolytic cleavage of Notch1, which leads to increases in NICD production. In addition to this post-translational effect, some studies suggest that neural activity also impacts Notch signaling at the transcript level of *Notch1* mRNA [127]. Hence, multiple mechanisms may act in concert to fine-tune the amount of Notch signaling that is induced upon neural activation. Neuronal activity-dependent Notch1 activation has also been found in the mouse olfactory system [128]. Unlike in *Drosophila* in which Notch expression and activation were found mostly in the primary ORNs, odor-dependent Notch1 activation was seen in mitral cells, which are secondary neurons that functionally correspond to projection neurons in *Drosophila*. Hence, although Notch activation is seen in the olfactory systems of both flies and mice, their precise role in olfaction may have diverged over evolution. Neuronal activity-dependent Notch signaling has also been reported in regions of the brain that comprise the hypothalamic-neurohypophysial system [129]. Notch3 and Dll4 are the key receptor and ligand that are implicated in this context, indicating that different regions of the mammalian brain may activate Notch signaling via different receptors and ligands in a neuronal activity-dependent manner.

### 3.2. Post-Developmental Notch Signaling Regulates Neural and Synaptic Physiology

While neuronal activity can trigger or modulate Notch signaling, Notch signaling can also reciprocally alter neuronal physiology and synaptic activity post-developmentally. The first direct implication of Notch signaling in synaptic physiology came from experiments performed in mice [130]. By studying a transgenic mouse line that expresses an antisense RNA that causes a 40–50% decrease in Notch1 protein levels in the hippocampus [130], Wang et al. found that CA1 synapses of this brain region showed impaired LTP (long term potentiation) and enhanced LTD (long term depression) [130]. LTP and LTD are forms of synaptic plasticity that strengthen or weaken specific neuronal connections, respectively, and, thus, modulate neural circuit activity. In addition, the authors showed incubation of hippocampal slices with Jag1 peptides that can activate Notch signaling in situ and can enhance LTP in wild-type brains. Furthermore, they showed that this manipulation can partially suppress the LTP defect in the *Notch1* knock-down animals. Although the knock-down of *Notch1* performed in this original work was not strictly limited to post-developmental brain tissue, additional studies performed by Alberi et al. using a *CamKII-Cre* mediated *Notch1* cKO line further validated the role of Notch1 in synaptic plasticity in CA1 neurons [126]. Several studies using mice have also implicated additional components of Notch signaling in synaptic plasticity in the hippocampus including *Rbpj* [131], *Mind bomb-1* [103], and *Mind bomb-2* [105]. Considering that Jag1 is found in proximity to Notch1 in CA1 neurons [126] and *Mind bomb* genes encode E3 ligases that facilitate ligand-mediated Notch activation, it is debatable that canonical Notch signaling mediated by Jag1 and Notch1 regulates synaptic plasticity in this context. The function of Notch activity in modulating synaptic plasticity has also been examined in the visual cortex through gain-of-function studies [132]. By postnatally over-expressing NICD of Notch1 in cortical neurons using the Cre/LoxP system during the critical period of ocular dominance, Dahlhaus et al. showed that induction of Notch signaling in V1 pyramidal neurons reduces LTP in this context. This result is in contrast with the hippocampal CA1 synapses that showed an increase in LTP upon Notch signaling activation, which suggests context or the method utilized to activate Notch signaling (NICD over-expression versus ligand-like peptide application) may be important.

What are the molecular mechanisms that underlie the changes in synaptic plasticity upon manipulation of Notch signaling in post-developmental brains? In addition to the potential role of Notch signaling in regulating CREB signaling [51] and cell-cell adhesion [61] as discussed earlier, Notch signaling may also mediate changes in synaptic physiology by modulating Reelin, glutamate, and the GABA (Gamma-AminoButyric Acid) signaling pathways [59,131].

Brai et al. identified that Notch1 co-localizes with components of the Reelin signaling pathway in hippocampal neurons using confocal and electron microscopy techniques [59]. Notch1 was found to be enriched in post-synaptic areas in co-localization with NMDA (N-methyl-D-aspartate) receptors, and was also found together with a Reelin receptor ApoER2 (Apolipoprotein E Receptor 2) and a downstream Reeling signaling molecule Dab1 (Disabled 1). Reelin signaling modulates synaptic plasticity as well as neural development by altering gene expression as well as cytoskeletal networks [133]. Another mechanism by which Reelin signaling modulates synaptic plasticity is through changes in NDMA receptor composition [134]. Most NMDA receptors are composed of two obligate GluN1 (a.k.a. NR1) subunits and two GluN2 (a.k.a NR2) subunits. Biochemical and electrophysiological properties of NMDA receptors are primarily determined by the GluN2 subtypes (e.g., GluN2A and GluN2B) which are being integrated into the tetrameric complex [135]. In *Notch1* cKO hippocampal neurons, the expression of GluN1 and GluN2B were found to be significantly decreased, while expression of GluN2A and AMPA (α-amino-3-hydroxy-5-methyl-4-isoxazolepropionic acid) receptor subunits GluR1 and GluR2 were unaltered [59]. Furthermore, the authors found that Notch1 and GluN1 can physically interact based on co-immunoprecipitation experiments, which suggests that these proteins may also functionally interact. Links between Notch and glutamate signaling have also been suggested in the mouse cerebellum [136,137] and rat hippocampus [138] as well as in *Drosophila* [61,64,139] and *C. elegans* [140] nervous systems. However, the proposed mechanism of action is different in different studies. For example, one study suggested that the expression of a vesicular glutamate transporter VGLUT1 (Vesicular GLUtamate Transporter 1) as well as other synaptic proteins such as Synaptophysin 1 are positively regulated by a non-canonical mode of Notch signaling that is ligand-dependent but γ-secretase independent [141]. In summary, although there seems to be many interesting links between Notch signaling and Glutamate signaling, how these pathways interact may be highly context-specific.

Another neurotransmitter that links Notch signaling to synaptic plasticity is GABA. Liu et al. identified that GABA transporters are target genes of Notch1-Rbpj signaling in the mouse hippocampus based on microarray and chromatin immunoprecipitation (ChIP) analyses [131]. Expression of GABA transporters, *Slc6a12* (i.e., *Bgt1*), and *Slc6a13* (i.e., *Gat2*) are significantly down regulated in the absence of *Notch1* in CA1 neurons, which suggests that Notch signaling may reduce extracellular GABA levels by upregulating the transporters that uptake them [131]. The authors hypothesized that this leads to a reduction in the inhibitory GABA signaling, which in turn leads to the LTP and LTD defects observed in *Notch1* and *Rbpj* cKO mice. Regulation of GABA signaling via Notch signaling has also been reported in *C. elegans* [142] and zebrafish [143], which suggests the cross-talk of these two pathways may mediate synaptic plasticity in multiple species in different contexts.

In summary, Notch signaling likely regulates synaptic plasticity through multiple molecular and cellular mechanisms, which requires further investigations in diverse neural circuits and organisms. Since Notch signaling is highly context-dependent, it is very dangerous to assume that a mechanism discovered in one system can be directly applied to a similar system in the same or different species.

### 3.3. Post-Developmental Notch Signaling Regulates Neuronal Morphology

Alterations in neural physiology are often accompanied by changes in neuronal and/or synaptic structure. During development, Notch receptors can regulate gross neuronal morphology and fine synaptic architecture through both canonical and non-canonical signaling in diverse species. This includes regulation of neuronal migration [144,145,146,147], neurite outgrowth and morphogenesis [148,149,150,151,152,153,154,155,156,157,158], axon guidance and targeting [159,160,161,162,163,164], dendrite morphogenesis [165,166,167,168,169,170,171,172], synaptogenesis [173,174], and synaptic growth [121,175]. Hence, one may consider that Notch signaling may regulate neuronal and synaptic physiology by fine-tuning cellular mechanisms that lead to structural alterations in the post-developmental brain.

Several studies using fruit flies have revealed the post-developmental role of Notch signaling in synaptic morphology. In the *Drosophila* olfactory system, Notch signaling regulates the size of a glomerulus, which is a synaptic structure formed by primary ORNs that expresses a certain odorant receptor and corresponding second order post-synaptic neurons called projection neurons [176]. The mechanism by which Notch signaling regulates glomerulus size in this context seems to be complicated because both canonical as well as non-canonical modes of Notch signaling both contribute to this phenomena [176,177]. When flies are first exposed to a specific odor, non-canonical Notch signaling, which does not require Notch ligands or proteolytic cleavage of the receptor, promotes the size increase of a glomerulus that is innervated by ORNs that respond to this specific odor. This non-canonical mode of Notch signaling seems to involve the Abl tyrosine kinase pathway [176], which is a non-receptor kinase cascade that has been linked to non-canonical Notch signaling during axon guidance in *Drosophila* [178,179]. Upon chronic exposure to the same odorant, the Notch ligand Delta expressed in post-synaptic projection neurons activates Notch signaling in ORNs. This canonical form of Notch signaling, in turn, restricts the growth of the glomerulus. Importantly, this canonical activation of Notch also results in the changes of neuronal physiology and synaptic connectivity between the ORNs and projection neurons [176], whereas morphological alternations mediated by non-canonical Notch signaling can be uncoupled from functional changes in this neuronal network [177]. Therefore, even though Notch signaling can induce morphological changes in neuronal and synaptic architecture, one must not assume that this is the underlying cause of the functional alterations that takes place in specific neuronal populations upon manipulating Notch signaling in vitro and in vivo.

## 4. Notch Signaling in Neuronal Maintenance

In addition to the diverse roles that Notch signaling plays in neural physiology and behavior, this pathway has also been studied in the context of neural maintenance and degeneration [34]. One of the strongest motivations to study Notch signaling in these contexts is that rare mutations in genes that encode the catalytic subunits of the γ-secretase complex, *Presenilin 1* (*PSEN1*) and *PSEN2*, cause rare early-onset familial Alzheimer’s disease [180,181]. More recently, genetic variations in *ADAM10* [*kuzbanian (kuz)* in *Drosophila*), a protease required for proteolytic cleavage of Notch receptors prior to γ-secretase-dependent cleavage, and *TM2D3*, a regulator of Notch signaling that has been genetically linked to γ-secretase activity, have been identified as susceptibility loci for the more common late-onset AD through genome-wide and exome-wide association studies [182,183]. Since ADAM10 and γ-secretase are also involved in the proteolytic cleavage of the Amyloid Precursor Protein (APP), which is encoded by another early-onset AD associated gene, determining whether Notch signaling defects contribute to AD pathogenesis has been an important topic in the AD research field. We will discuss the role of Notch signaling in AD in more depth in the next section (5.1). In addition, rare variants in *NOTCH3* are linked to a major hereditary vascular neurodegenerative disease called CADASIL (Cerebral Autosomal Dominant Arteriopathy with Subcortical Infarcts and Leukoencephalopathy) [184]. Although the neurodegenerative phenotype in CADASIL patients is considered to be a secondary consequence of defects in arterial blood vessel integrity [185] and *NOTCH3* is primarily expressed in vascular smooth muscle cells [186], researchers have been keen to understand whether defects in Notch signaling in neurons may contribute to the pathogenesis of this and other neurodegenerative diseases.

Although the possibility of Notch signaling being involved in the maintenance of neuronal integrity has been discussed in a number of research and review articles [187,188,189], most arguments are based on indirect evidence. For example, Presente et al. reported that *N^ts1^* flies reared in restricted temperature after three days post-eclosion are short lived, and this phenotype can be rescued by reintroducing a wild-type copy of *Notch* provided as a chromosomal duplication or as a cDNA transgene [190]. These animals also exhibit age-dependent flight and climbing phenotypes, which suggests that loss of Notch signaling in adult flies leads to progressive motor function defects. The authors of this study hypothesized that the motor and lifespan defects observed in adult flies defective of Notch signaling are due to neurological issues because the Notch protein is more abundantly expressed in the adult head (mostly composed of neurons) when compared to the thorax (mostly made up of muscles). However, they were unable to observe any histological signs of neurodegeneration in *N^ts1^* flies based on silver-stained sections of the adult brain. In addition, neuron-specific manipulations of Notch (i.e., using RNAi or dominant-negative Notch transgenes) were not performed in this study. Another group also reported that *N^ts1^* flies that shifted to restrictive temperature post-developmentally became flightless within a few days, and showed that these animals exhibited gross morphological changes in flight muscles based on transmission electron microscopy [191]. In addition, recent studies have revealed a role of Notch signaling in maintaining the homeostasis of adult muscles [192], digestive system [193], and reproductive organs [194], which suggested that defects in these and other adult organ systems may contribute to age-dependent phenotypes in *N^ts1^* flies.

Several lines of evidence in mouse models suggested that neither loss or gain of canonical Notch signaling in adult neurons is sufficient to trigger neurodegeneration. Although Notch1 expression [195] and its processing in the brain [196] has been found to decrease with age, cKO studies of *Notch1* and other genes in the pathway argue against the idea that Notch signaling is necessary for neuronal integrity in the aging brain. In 2012, Zheng et al. generated cKO animals for *Notch1* and *Notch2* in the postnatal mouse forebrain using *CaMKII-Cre* [197]. To their surprise, single or double cKO mouse lines did not exhibit any histological signs of neurodegeneration in the cerebral cortex for up to about two years of life. Since *Notch3* and/or *Notch4* could be playing a redundant or a compensatory role in this context, another study generated *Rbpj* cKO mice using the same *CamKII-Cre* driver to fully inhibit canonical Notch signaling [66]. The authors examined these animals up to 18 months, and similar to the *Notch1/Notch2* double cKO animals, *Rbpj* cKO mice also did not show any signs of neurodegeneration. In addition to these loss-of-function studies, one group explored whether hyperactivation of canonical Notch signaling is sufficient to trigger neurodegeneration in post-developmental neurons [132]. The authors reported that they did not detect any histological signs of neurodegeneration when NICD of Notch1 is over-expressed in cortical pyramidal neurons in 2.5-month-old mice. Although this study did not follow their animals for a long time, it is unlikely that hyper-activation of Notch signaling is detrimental to neuronal health since ectopic NICD expression is usually considered a very strong manipulation to activate the canonical signaling pathway in vitro and in vivo. Taken together, these studies suggest that removal or over-activation of the canonical Notch signaling pathway within the nervous system in a non-disease state appears to have no major detrimental effect to the health of neurons.

## 5. Notch Signaling as a Modifier of Neurodegenerative Diseases (Table 3)

Although studies of both LOF and gain of function (GOF) of Notch signaling suggest that too little or too much Notch signaling in post-developmental neurons is unlikely to be sufficient to cause a strong neurodegenerative phenotype, it is still possible that alterations in Notch signaling can modulate the neurodegenerative phenotypes that are triggered by other genetic and/or environmental factors. In this section, we will introduce and discuss studies that have implicated Notch signaling in the pathogenesis of several neurodegenerative diseases using fly or mouse models. *Drosophila* is an excellent model system to explore this link because genetic enhancer and suppressor screens can be easily conducted using neurodegenerative disease model flies that ectopically over-express pathogenic human proteins in vivo in a tissue and cell type specific manner [198]. In addition, flies can be used to study the functional impact of genetic variants of unknown significance in Notch-related genes to assess whether rare missense variants associated with human diseases may alter protein function [10,199]. Once a strong functional link is made between a genetic variant and a neurodegenerative disorder, flies can be further used to dissect the molecular mechanisms that underlie the pathogenesis [200,201]. Although most of the data discussed in the following section require further validations, especially in humans, they illustrate how insights from genetic model organisms allow the formulation of new hypotheses and theories that can be tested through further translational and clinical studies [202].

### 5.1. Alzheimer’s Disease (AD) and Related Dementia

AD is a neurodegenerative disease primarily characterized by dementia [203]. AD can be grossly categorized into early-onset and late-onset subtypes depending on the age of onset (before or after 65 years old, respectively). Mutations in the genes *PSEN1* and *PSEN2* have been shown to cause early-onset familial forms of the disease [180,181], and the encoded Presenilin proteins act as the catalytic subunit of γ-secretase, which generates the characteristic amyloid plaques found in AD brains through proteolytic cleavage of APP. While the *PSEN* mutations are believed to cause an increase in the amount of a toxic form of β-amyloid (Aβ42) [204], it has also been shown that PSENs and γ-secretase are required for neuronal maintenance independent of its function on APP. Experiments in mice have shown that post-natal knockout of *Presenilin* genes (*Psen1* and *Psen2*) in the forebrain causes impairments in synaptic plasticity and age-dependent neurodegeneration [205]. Additionally, the removal of *Nicastrin* (encoding another subunit of γ-secretase) in the forebrain also causes memory impairment, neurodegeneration, and progressive gliosis in mice [206]. Analogous experiments performed in *Drosophila* in which *Presinilin (Psn)* or *Nicastrin (Nct)* expression were specifically reduced in neurons, which also resulted in age-dependent neurodegeneration [207]. This indicated that this neuroprotective function of the γ-secretase complex is evolutionarily conserved.

As mentioned previously, Notch signaling relies on Presenilins/γ-secretase for signal transduction. Given the requirement for γ-secretase activity for neuronal survival, it is reasonable to hypothesize that aberrant Notch signaling due to disturbed activity of γ-secretase may modulate/contribute to neurodegeneration in AD. A *PSEN1* mutation found in patients with familial frontotemporal dementia has also been found to inhibit cleavage of APP and Notch by γ-secretase [208]. Moreover, a rare deleterious variant in the human gene *TM2D3* was found to have a strong association with increased risk of late-onset AD [182]. Studies in *Drosophila* have shown that the mutants of the *TM2D3* ortholog, *almondex* (*amx*), show strong maternal-effect Notch LOF phenotypes [209,210]. In addition, epistasis experiments have suggested that *amx* likely regulates the function of γ-secretase [211]. Several papers also suggested that APP and Notch receptors can physically interact with one another [212,213,214]. However, the pathophysiological significance of this interaction is unknown. Lastly, chronic over-expression of the NICD in a transgenic rat model of AD over-expressing Aβ42 results in exacerbation of spatial memory defects [215]. These data highlight the need for proper γ-secretase function in neuronal health and suggest that abrogated Notch signaling may play a modulatory or causative role in neurodegeneration in the context of AD and related dementia.

### 5.2. Huntington’s Disease (HD)

HD is a neurodegenerative disease characterized by uncontrolled movements, cognitive decline, and emotional issues [216]. HD is caused by poly-glutamine (polyQ) expansions in the *Huntingtin* (*HTT*) gene. Using a fly model of HD in which mutant Huntingtin protein was overexpressed in the eye causing an age-dependent rough eye phenotype, Calpena et al. found that knock-down of *junctophilin* (*jp*) enhances this phenotype while overexpression of *jp* suppresses this defect [217]. In the same paper, the authors noted that ubiquitous knock-down of *jp* causes bristle and wing vein phenotypes that are reminiscent of Notch LOF defects, and further showed that *jp* genetically interacts with *Delta* mutants. Junctophilins are a conserved family of proteins found in excitable cells, and they are considered to modulate calcium signaling in these cells [218]. Mammals have four genes (*JPH1-4* in humans) that correspond to a single *jp* gene in *Drosophila*, and mutations in *JPH3* are known to cause a Huntington-like disease (OMIM #606438). Considering that calcium signaling also impacts Notch signaling [219], it is interesting to speculate that *jp* may be modulating the toxicity of mutant HTT by modulating Notch signaling.

Postmortem brains of HD patients have been shown to have alterations in gene expression including some that correlate with the severity of disease pathology. In this context, the Notch target gene *HES4* (encoding a bHLH transcription factor, *hairly* (*h*) in *Drosophila*] was reported to show increased DNA methylation at its locus that correlates with the severity of striatal degeneration [220]. The authors also observed a reduction in the expression of *HES4* target genes, including *MASH1* (a.k.a. *ASCL1*) and *P21* (a.k.a. *CDKN1A*) that were implicated in striatal GABAergic neurodevelopment. The study further showed that knockdown of *HES4* was able to suppress cell death caused by mutant HTT over-expression in a neuroblastoma cell line. This effect was also accompanied by reduction of mutant HTT aggregate formation, which indicated that HES4 and potentially Notch signaling may alter protein homeostasis of mutant HTT. Given that development of HD is a lifelong process, disruption of Notch signaling in the developing brain in addition to alterations in its post-developmental function may both contribute to disease pathogenesis of HD.

### 5.3. Spinocerebellar Ataxia (SCA)

SCA is a disorder characterized by progressive degeneration of brain regions that regulate movement control [221]. SCA can be classified into different subtypes primarily based on the genes that are mutated and types of mutations that are found in the patient [222]. Similar to HD, many SCAs are caused by poly-Q expansions in certain genes. In addition to toxicity caused by poly-Q tracts that cause cellular stress, modulation of the host protein function by poly-Q stretches are also thought to contribute to disease pathogenesis. SCA1 is caused by poly-Q expansion in *Ataxin-1* (*ATXN1* in human, *Atx-1* in *Drosophila*), which encodes a nuclear protein enriched in Purkinje neurons of the cerebellum [223]. In an effort to understand the endogenous function of wild-type ATXN1 without the poly-Q expansion, Tong et al. over-expressed this protein in *Drosophila* [224]. Expression of ATXN1 in the developing larval wing disc caused cell lethality whereas expression of a related protein, ATXN1L (a.k.a. *Brother of Ataxin-1*, also orthologous to *Atx-1* in flies), resulted in wing vein thickening and reduced expression of *E(spl)mβ*, which is a Notch target gene. These Notch-related phenotypes caused by ATXN1L can be suppressed by co-expression of Su(H). Further experiments in mammalian cells identified that ATXN1 and ATXN1L can bind to and inhibit RBPJ, which suggested that these proteins can regulate canonical Notch signaling. Further studies are required to determine whether functional alterations in this Notch modulating function of ATXN1 contributes to the pathogenesis of SCA1.

Another link between Notch signaling and SCA has been suggested using a fly model for SCA17, which is a SCA subtype caused by poly-Q expansions in *TATA-box-binding protein* (*TBP*). Expression of disease associated form of TBP with 80 poly-Q expansions (TBP80Q) causes motor defects and a shortened lifespan phenotype when expressed in the central nervous system in *Drosophila* [225]. This pathogenic protein also mediates progressive retinal degeneration in the fly eye using an eye-specific driver. In a screen to identify genetic modifiers of this eye phenotype, the authors found that overexpression of *Su(H)* can suppress the TBP80Q-induced rough eye phenotype, while RNAi against *Su(H)* enhances this defect [225]. Additional experiments showed that Su(H) can bind to pathogenic TBP80Q but not to normal TBP and that ~40% of genes dysregulated by TBP80Q overexpression have at least one upstream Su(H) binding site in their respective genomic locus. These data suggest that, in patients with SCA17, TBP with expanded poly-Q tracts may cause a widespread disruption of proper Notch signaling by sequestering RBPJ or by disrupting its function. Considering that several additional studies in *Drosophila* [226], mouse [227], and cell-based systems [228] also suggest molecular links between genes associated with SCA and Notch signaling, further functional studies will likely reveal whether and how defects in Notch signaling contribute to the pathogenesis of this movement disorder.

### 5.4. Amyotrophic Lateral Sclerosis (ALS)

ALS is a neurodegenerative disorder that affects the motor system [229]. Mutations in genes including *C9ORF72* (*Chromosome 9 Open Reading Frame 72*), *SOD1* (*SuperOxide Dismutase 1*), and *TARDBP* (*TAR DNA binding protein*, also known as *TDP-43*) have been linked to this condition [230]. Studies in *Drosophila* revealed that dipeptide repeat causing mutations in *C9ORF72* associated with ALS produce abnormal protein products through repeat associated non-AUG (RAN) translation that can inhibit Notch signaling in vivo [231]. In the same study, the authors confirmed that some Notch target genes were downregulated in postmortem ALS patient brains as well as in iPSC (induced Pluripotent Stem Cell)-derived cortical neurons. Another work identified that a candidate peptide drug (GM604) that has been developed for ALS treatment can activate Notch signaling along with other pathways when tested in a human cell line [232], which indicates that activation of Notch signaling may have beneficial effects on neuronal survival. In contrast to these studies, however, a *Drosophila* paper focusing on *TDP-43* suggested that increased Notch signaling may contribute to ALS pathogenesis by performing transcriptomic analysis and genetic interaction experiments on flies over-expressing a mutant form of TDP-43 [233]. Upregulation of Notch signaling components and hyperactivation of this pathway were also reported in two studies using in vivo mouse and in vitro human cell-based assays [234,235]. The authors of one study reported that expression of all four NOTCH receptors are increased in spinal cords of ALS patients, along with increased expression of the Notch ligand *JAG1* in motor neurons and reactive astrocytes [235]. In the same paper, *Jag1* deletion was found to aggravate disease progression in an ALS mouse model that over-expresses mutant *SOD1*. Although these studies suggest that the Notch pathway is likely aberrantly regulated in ALS patient brains and manipulations of the pathway can modulate disease-related phenotypes in multiple model systems, additional studies are required to determine whether Notch signaling aggravates or suppresses disease progression before translating these findings to clinical care.

### 5.5. Parkinson’s Disease (PD)

PD is characterized by motor symptoms accompanied by the degeneration of dopaminergic neurons in the substantia nigra [236]. Mutations in genes such as *SNCA* (SyNuClein Alpha, encoding α-Synuclein protein), *LRRK2* (Leucine Rich Repeat Kinase 2), and *VPS35* (Vacuolar Protein Sorting 35 retromer complex component) are associated with this condition [237]. Expression of α-Synuclein has been shown to alter Notch1 expression and subsequent Notch signaling in mice as well as in mouse embryonic stem cells [238]. Furthermore, fly orthologs of *LRRK2* (*Lrrk*) and *VPS35* (*Vps35*) has been shown to modulate Notch signaling by altering the endolysosomal trafficking pathway in vivo in *Drosophila* as well as in other experimental settings [239,240]. Although the precise role of Notch signaling in PD is still obscure, further functional characterization of genes involved in PD in the context of Notch signaling as well as exploration of Notch signaling activity in brains of PD patients will likely clarify this relationship.

### 5.6. Prion Diseases

Prion diseases are a group of disorders in which transmittable neurotoxic proteins cause neurodegenerative phenotypes [241]. Mutant forms of *PRNP* (*PRioN Protein*), which is designated as PrP^Sc^, can cause a wide range of prion diseases in multiple mammalian species. This includes Creutzfeldt-Jakob disease in human, Scrapie in sheep, and bovine spongiform encephalopathy in cattle. In one study, PrP^Sc^ was found to increase the expression of *Notch1* mRNA as well as facilitate the production of NICD in mice [242]. This activation of Notch1 correlated with the amount of prion accumulation in synaptosomes, and was accompanied with dendritic atrophy in PrP^Sc^ expressing neurons. Inhibition of Notch1 promoted dendritic growth in this context, which suggested that PrP^Sc^ may facilitate Notch1 expression and signaling to somehow promote dendritic atrophy. Hence, manipulations to inhibit Notch signaling may be useful in preventing some negative effects of prion proteins on neuronal function.

### 5.7. Multiple Sclerosis (MS) and Myelination-Related Disorders

MS is characterized by demyelination of axons in the brain and spinal cord [243]. Pathogenesis of MS involves complex interplay between neurons, glial cells, and the immune system. Although many agree that Notch signaling is involved in the processes of demyelination/remyelination in mammals, the precise role of Notch in these processes and in MS is still a subject of debate.

Some studies in rats have shown that Notch acts as a positive regulator of remyelination [244,245]. Notch activation was found in neural progenitor and oligodendrocyte progenitor cells (OPCs) after toxin-induced demyelination, and γ-secretase inhibition was shown to block this process. Moreover, after demyelination, *Hes5* is triggered to promote OPC proliferation, which is essential for remyelination of neurons [245]. Notch is also involved in the maturation of oligodendrocytes and myelination during development, which is a context in which Contactin1 (a.k.a. F3/contactin), a GPI-linked neural cell recognition molecule, serves as a functional ligand for the Notch receptor [244,246]. While the positive role of Notch in remyelination has been shown in these and other studies, some argue that Notch can function as a negative regulator of remyelination, or is not even involved in this process. For example, Endothelin-1 (ET-1) expressed in astrocytes was shown to slow the rate of remyelination by promoting Jag1 expression in astrocytes to further induce Notch activation in OPCs in mice [247]. In another study, inhibition of *Notch1* through siRNA was shown to accelerate remyelination in a mouse model of MS induced by cuprizone [248]. In contrast to these studies, one group argued that, despite the expression of Notch1 in OPCs upon demyelination, Notch signaling neither plays a crucial role on the rate of remyelination or prevents remyelination [249]. To further complicate the issues, Notch signaling also plays complex regulatory roles in immune cells during the pathogenesis of MS and other myelination-related autoimmune disorders [250,251,252,253,254,255,256,257,258]. Considering that Notch is a highly context-specific pathway that can inhibit or activate distinct sets of genes in diverse contexts, further dissection of the role of this pathway in different cell populations using sophisticated genetic tools will be critical to understand its role in MS and other related disorders.

### 5.8. Neurotoxic Environmental Factors

In addition to potential links between Notch signaling and neurodegeneration caused by genetic factors, Notch signaling has been proposed to be involved in mediating the death of neurons caused by some neurotoxic chemicals. For example, the excitotoxic chemical, kainic acid, has been shown to be damaging to neurons by inducing ectopic G-to-S phase transition in post-mitotic neurons [259]. The authors of this study found that this effect is mediated by ectopic activation of Notch signaling, which, in turn, promotes apoptosis in mature neurons. This cell death triggered by kainic acid can be suppressed by the loss of *Rbpj*, which indicated that canonical Notch signaling is responsible for neuronal cell death caused by ectopic cell cycle entry in post-mitotic neurons.

Another environmental toxin that has been thought to mediate some of its actions through Notch signaling is methylmercury (MeHg). In a pioneering study, a cell-based study in *Drosophila* showed that exposure to MeHg activates several Notch signaling target genes in the *E(spl)* complex in a Notch receptor-dependent manner [260]. The same group also subsequently reported that MeHg can activate some Notch signaling target genes in a Notch receptor and Su(H)-independent manner in *Drosophila* cells in culture as well as in vivo [261,262], which suggests that MeHg can regulate multiple Notch downstream target genes in different ways. In addition, this effect is likely to be evolutionarily conserved since MeHg was also shown to positively affect Notch signaling using rat embryonic neural stem cells [263]. It is worth noting, however, that another group proposed MeHg can silence Notch signaling by directly inhibiting γ-secretase mediated processing of Notch receptors by using cellular assays and in vivo fly models [264]. Hence, additional studies in diverse contexts will be required to elucidate the precise role for MeHg in Notch signaling to determine whether aberration of Notch signaling contributes to neurotoxic effects of this molecule.

In addition to kainic acid and MeHg, Notch signaling has been shown to be affected by other chemicals that have been linked to neurological disorders including polychlorinated biphenyls [265] and nanoparticles [266]. Notch signaling also seems to be relevant to neuronal death following trophic factor withdrawal [267] and hyperoxia-induced brain damage [268]. Hence, in addition to being implicated in diverse neurodevelopmental, neurological, and psychiatric diseases caused by genetic factors, Notch signaling is also likely to contribute to conditions that are caused by environmental factors or through gene-environment interactions. We envision that studies on the role of Notch signaling in infectious diseases that impact the developing and post-developmental nervous systems (e.g., Zika virus) will also be a topic of great interest [269,270].

## 6. Conclusions and Future Directions

Based on the manuscripts that we discussed in this review article, we conclude that Notch signaling plays numerous context-specific roles in the post-developmental nervous system in diverse species. First, Notch signaling plays critical roles in learning and memory in multiple species, but their precise function can be species-specific, which highlights the importance of performing experiments in multiple organisms in vivo. Second, Notch signaling is involved in a number of additional complex behaviors including addiction, sleep, fear, anxiety, pain perception, and sociability, which suggests that this pathway may be relevant to psychiatric disorders in humans. Third, Notch signaling is regulated by neuronal activity in certain contexts, and activation of this pathway can, in turn, modulate neuronal function through different mechanisms including alteration of synaptic plasticity and neuronal morphology. Fourth, although Notch signaling is clearly required for various aspects of neural physiology, this pathway does not seem to have an essential role in neuronal survival. Fifth, studies of different neurodegenerative disease models and environmental factor exposure suggest that alteration of Notch signaling likely contributes to certain aspects of these neurological disorders, which indicates that modulation of Notch activity may be effective in treating some symptoms associated with these conditions.

While the canonical Notch pathway is seemingly simple and straightforward, the pathway is highly pleiotropic and context-specific. There are many downstream target genes that are activated in some but not in other contexts, and many of these targets encode transcription factors that, in turn, regulate a host of other genes. Moreover, because most of the studies discussed in this paper did not shed light on what the precise downstream effectors of Notch signaling are in each context and how exactly they may be modulating the disease phenotypes, further mechanistic studies are needed to fully understand how Notch may modulate and impact different neurodegenerative diseases. Considering the numerous cell biological processes and downstream target genes Notch signaling can impact, some of which are mediated by non-canonical modes of signaling, it is unlikely that there is a single common mechanism by which Notch signaling modifies symptoms seen in neurodegenerative disease patients.

Since Notch signaling is reiteratively used during development and post-development in different tissues, cell types, and contexts, sophisticated genetic manipulation is required to manipulate Notch signaling in a highly spatially and temporally coordinated manner to understand the precise role of this pathway in the post-developmental nervous system. Model organisms such as *Drosophila* and *C. elegans* are highly suited for these types of analyses, and additional studies from these and other fields are likely to expand our understanding of the precise role Notch signaling has in post-developmental nervous systems. The studies from mammalian model organisms such as mouse and rat models will further provide a more comprehensive view that will allow researchers to compare and contrast the roles of Notch signaling in adult brains of vertebrates and invertebrates, especially in the context of adult neurogenesis and the adaptive immune system. Studies based on human subjects will complement research performed in model organisms allowing the translation of knowledge derived from genetic model organisms to personalized care.

## Figures and Tables

**Figure 1 biomolecules-10-00985-f001:**
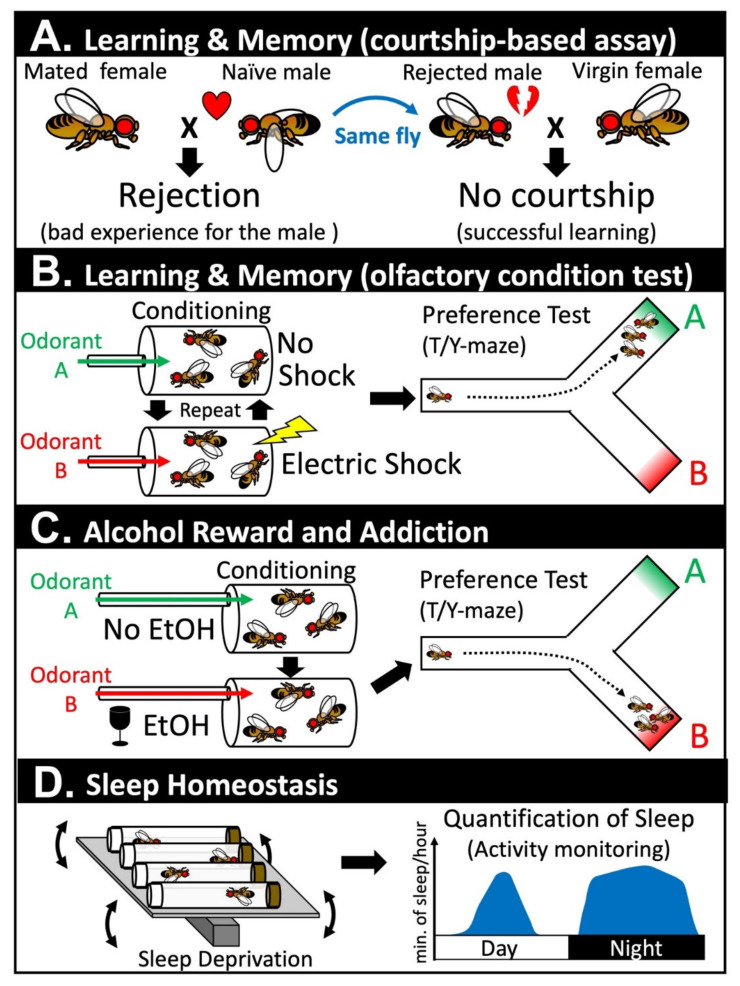
Assays in *Drosophila* that have been used to assess post-developmental functions of Notch signaling in the adult nervous system. (**A**) Learning and memory can be assessed via a courtship-based assay. Naïve male flies have a natural instinct to court female flies. A male will be successful if he courts a virgin female, but will be strongly rejected if he courts a mated female. Once rejected, the male fly learns from this unpleasant experience and will not attempt to court another female for a period of time even when paired with a receptive virgin female. (**B**) Learning and memory can also be assessed using an olfactory condition test. In this assay, a group of flies are presented with a neutral odorant (odorant A) and an odorant that is paired with an electric shock (odorant B). After repeating this training several times, the flies are then put into a T-maze or a Y-maze to assess whether they learned to avoid odorant B. These flies can be collected again and be further tested at a later timepoint to assess their ability to retain their memory. (**C**) Response to alcohol reward and addiction can be assessed by exposing a group of flies to a neutral order (odorant A) or an odorant that is paired with vaporized ethanol (EtOH). The flies are then put into a T/Y-maze to assess whether they established a preference toward odorant B. Flies that are addicted to EtOH show further alcohol seeking behavior even under a condition in which flies need to tolerate punishment (electric shocks) to receive the reward (not depicted here). (**D**) Sleep homeostasis can be assessed by, first, sleep deprivation of a group of flies using a ‘Sleep Nullifying Apparatus’ that continues to tilt and rock small vials that house these animals. These sleep-deprived flies can be assessed for their quantity of sleep using a ‘*Drosophila* Activity Monitor.’ A bout of sleep in *Drosophila* is defined as a period of immobility that lasts for 5 min or more. Flies that have a normal homeostatic response will sleep more after sleep deprivation, whereas flies that are defective do not show this rebound.

**Table 1 biomolecules-10-00985-t001:** Behaviors that are affected by post-developmental Notch signaling.

**Learning and Memory**				
**Models**	**Genes**	**Alleles**	**Phenotypes**	
*Drosophila*	*N*	*N^ts1^* and *N^ts2^* (temperature-sensitive LOF), RNAi, and NΔcdc10rpts (dominant negative)	Disruption of long-term memory formation.	(Ge et al., 2004, Presente et al., 2004)
*Drosophila*	*N*	*N^nd-1^* and *N^nd-3^* (temperature-sensitive GOF)	Upregulation of a long-term memory (LTM) mediator, CREB (cAMP response element binding protein).	(Zhang et al., 2013)
*Drosophila*	*N*	*hs-NICD* (overexpression of active form of Notch)	Disruption of long-term memory formation.	(Zhang et al., 2015)
*Drosophila*	*Su(H)*	*Su(H)^IB115^* and *Su(H)^HG36^* (LOF), and *hs-Su(H)* (Overexpression)	Disruption of long-term memory formation.	(Song et al., 2009)
*Drosophila*	*klg*	*klg^rus^* (LOF)	Impairment of Notch-dependent long-term memory formation.	(Matsuno et al., 2009)
Mouse	*Jag1*	Heterozygous null mutants	Impairment of spatial memory formation.	(Sargin et al., 2013)
Mouse	*Adam10*	*Adam10^flox/flox^* conditional KO	Impairment of learning and memory in the Morris water-maze.	(Zhuang et al., 2015)
Mouse	*Rbpj*	*Rbpj^flox/flox^* conditional KO	Defects in long-term potentiation and in learning and memory.	(Liu et al., 2015)
**Models**	**Genes**		**Key findings**	
Rat	*Notch2*		High expression of Notch2 in hippocampus and cerebellum upon establishment of long-term spatial memory.	(Storozheva et al., 2017)
**Reward and Addiction**				
**Models**	**Genes**	**Alleles**	**Phenotypes**	
*Drosophila*	*sca*	*sca^5-120^* and *sca^BP-2^* (LOF)	Impairment of reward memory of alcohol.	(Kaun et al., 2011)
*Drosophila*	*sca*	*sca^5-120^* and *sca^BP-2^* (LOF) and RNAi	Reduced Notch activation and reward memory of alcohol.	(Petruccelli et al., 2018)
*Drosophila*	*N*	RNAi	Impairment of alcohol associative preference.	(Petruccelli et al., 2018)
*Drosophila*	*Su(H)*	RNAi	Impairment of alcohol associative preference.	(Petruccelli et al., 2018)
**Sleep homeostasis**				
**Models**	**Genes**	**Alleles**	**Phenotypes**	
*Drosophila*	*N*	*N^spl-1^* (GOF)	Disruption of sleeping homeostasis after sleep deprivation.	(Seugnet et al., 2011)
*Drosophila*	*Dl*	Overexpression	Disruption of sleeping homeostasis after sleep deprivation.	(Seugnet et al., 2011)
*Drosophila*	*bun*	*bun^BG0162^*, *bun^KG06590^*, *bun^KG00456^*, *bun^KG00392^* (LOF) and *Df(2L)prd1.7* (Deficiency)	Defects in sleep rebound after sleep deprivation.	(Seugnet et al., 2011)
**Models**			**Key Findings**	
*C. elegans*	*osm-7, osm-11, lin-12, glp-1*		Notch signaling induces a lethargus-like quiescence state in adult worms and regulates developmental lethargus.	(Singh et al., 2011)

**Table 2 biomolecules-10-00985-t002:** Links between post-developmental Notch signaling and neural activity.

**Notch Signaling is Regulated by Neuronal Activity**			
**Models**	**Genes**	**Key Findings**	
*Drosophila*	*N*	Notch activation is induced in olfactory receptor neurons (ORN) in response to a selective stimulus in a Delta ligand-dependent fashion.	(Lieber et al., 2011)
Mouse	*Notch1*	Notch1 and Jag1 are found at the synapse in somatosensory cortex and hippocampus in a *Arc* (a.k.a. *Arg3.1*)-dependent manner. Neuronal activation triggers proteolytic cleavage of Notch1, leading to increases in Notch receptors (NICD).	(Alberi et al., 2011)
Mouse	*Notch1*	Neural activity-dependent alternative cleavage and polyadenylation impacts Notch signaling at the transcript level of Notch1 mRNA in hippocampus.	(Fontes et al., 2017)
Mouse	*Notch1*	Olfactory stimulation activates Notch activity in mitral cells of the mouse olfactory bulb.	(Brai et al., 2014)
Mouse	*Notch3* and *Dll4*	Neuronal activity-dependent reduction of DLL4 expression and proteolytic cleavage of Notch3 occur in the hypothalamic-neurohypophysial system.	(Mannari and Miyata, 2014)
**Post-Developmental Notch Signaling Regulates Neural and Synaptic Physiology**			
**Models**	**Genes**	**Key Findings**	
Mouse	*Notch1*	Expression of antisense RNA of Notch causes impaired LTP (long term potentiation) and enhanced LTD (long term depression) at hippocampal CA1 synapses.	(Wang et al., 2004; Alberi et al., 2011)
Mouse	*Mib-1*	Mib1-mediated Notch signaling controls synaptic plasticity and memory formation in hippocampus.	(Yoon et al., 2012)
Mouse	*Mib-2*	Mib2 regulates synaptic plasticity and spatial memory via the Notch signaling.	(Kim et al., 2015)
Mouse	*Notch1*	Postnatally overexpressed Notch1 signaling reduces LTP in the visual cortex.	(Dahlhaus et al., 2008)
Mouse	*Notch1*	Notch1 regulates the hippocampal synaptic plasticity through the interactions with the Reelin Pathway, glutamate, and CREB signaling pathways.	(Brai et al., 2015)
Mouse	*Notch1, Notch2, Jag1* and *Dll1*	Non-canonical Notch signaling positively regulates the expressions of VGLUT1 and Synaptophysin 1.	(Hayashi et al., 2016)
Mouse	*Notch1* and *Rbpj*	Notch1-Rbpj regulates the expression of GABA (Gamma-AminoButyric Acid) transporters such as Slc6a12 and Slc6a13 in CA1 neurons.	(Liu et al., 2015)
*C. elegans*	*lin-12*	LIN-12/Notch regulates synaptic activity by modulating GABA signaling at the neuromuscular junction.	(Sorkac et al., 2018)
**Post-Developmental Notch Signaling Regulates Neuronal Morphology**			
**Models**	**Genes**	**Key Findings**	
*Drosophila*	*N*	Non-canonical Notch promotes glomeruli volume increase. Then, canonical Notch regulates glomeruli volume and plasticity.	(Kidd et al., 2015; Kidd and Lieber, 2016)

**Table 3 biomolecules-10-00985-t003:** Notch-related genes studied in the context of neurodegenerative diseases.

Neurodegenerative Diseases	Relevant Human Genes Linked to Notch Signaling	Fly Homologs of Human Genes
Alzheimer’s disease (AD)	*PSEN1* *PSEN2* *NCT* *TM2D3* *ADAM10*	*Psn* *Psn* *Nct* *amx* *kuz*
Huntington’s disease (HD) and HD-like disease 2	*JPH3* *HES4*	*jp* *h*
Spinocerebellar ataxia 1 (SCA1)	*RBPJ* *HES6* *ATXN1* *ATXN1L (Brother of ATXN1*)	*Su(H)* *E(spl)m* *β* *Atx-1* *Atx-1*
Spinocerebellar ataxia 17 (SCA17)	*RBPJ*	*Su(H)*
Amyotrophic lateral sclerosis (ALS)	*NOTCH1* *NOTCH2* *NOTCH3* *NOTCH4* *JAG1*	*N* *N* *N* *N* *Ser*
Parkinson’s disease (PD)	*LRRK2* *VPS35*	*Lrrk* *Vps35*
Prion diseases	*NOTCH1*	*N*

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
