# Peer review of "Post-Developmental Roles of Notch Signaling in the Nervous System"

_biomolecules, 2020, doi:10.3390/biom10070985_

Round 1

Reviewer 1 Report

This is a timely, well-written and comprehensive review, dealing with the very important topic of the role of Notch signalling in the adult nervous system. Given the large number of investigators studying Notch signalling, adult neurophysiology and human nervous system disorders, this review will be of great interest for a broad audience. I have only a few suggestions for how to improve this piece.

1) Title and Abstract: Minor semantic issue, but maybe it would attract more readers if the phrase “post-developmental” was changed to simply “adult”.

2) Page 5, top: The heterozygous effects of Su(H) mutation (ref 50) could of course be due to developmental problems, and maybe this should be stated. The same applies, as far as I can see, to the description of some studies in mouse (ref 65, 66, 67; page 6).

3) The review is generally well-written. However, there are some grammatical errors throughout the piece.

4) Section 3, pages 8-11: They describe the adult role of Notch in “neural and synaptic physiology” (3.2) and “neuronal morphology” (3.3). But some of the adult roles of Notch may of course pertain to the role of Notch in adult neurogenesis. Maybe a section (3.4?) could be dedicated to this. As the authors point out, adult neurogenesis is poorly understood in flies, and likely does not occur outside of trauma, but it does occur in adult mammals. Hence, adult effects of Notch signalling may well emerge from adult neurogenesis effects.

5) Organisation: Section 4, on neuronal maintenance, deals with Notch in Alzheimers, but is followed by section 5, on neurodegenerative disease, which also deals with Alzheimers. Maybe change the section titles and/or move some paragraphs around to make it more logical.

6) Page 14, the second paragraph is in italics?

7) Conclusions and Future Directions (page 17): Worth mentioning again that siome of the adult roles of Notch, especially in mammals, may revolve around its known role in neurogenesis, and hence in adult neurogenesis.

Reviewer 2 Report

This presented work by Salazar and colleagues is comprehensive review about the post-developmental role of Notch signaling in the nervous system.  It is focused on work done with the model of Drosophila melanogaster but also includes studies done in C. elegans and vertebrate models. 

The selected studies reviewed are interesting to researchers working on brain function and neuronal degeneration under disease conditions.  The manuscript is well structured and easy to read. 

There are a few corrections and points I would like to propose:

Major points:

  1. Although the authors selected interesting and informative studies, the review is in parts a description of facts.  The author's efforts to discuss the selected studies and to make their own conclusion is limited, especially in chapter 5.  Could the authors discuss whether there are any common mechanism of how Notch misregulation leads to neurodegeneration.  What is known about the effector genes that are affected downstream of Notch? Are these genes coding for structural factors or genes coding for apoptosis factors?
  2. I think most readers would appreciate if the text would be illustrated with more figures.  In particular a figure about the connection between Notch, CREB and LTM would be helpful.  

Minor points:

  1. In the introduction the authors write of neural and dermal tissue.  I am not sure whether "dermal" is the right expression here for all the non-neural tissue.
  2. On page 5, it says Matsuno et al. tool a non-hypothesis driven approach [...]. It is not clear to me what "non-hypothesis" means in this context.
  3. On page 9, ...ligand that are involved in this contect, [...]
